# Reproducibility Report: Principle Feature Visualisation in Convolutional Neural Networks

1    **Reproducibility Summary**

2    Explainable AI is crucial in determining the performance of machine learning systems and debugging the code wherever
3    necessary. This motivated us to pursue the project for establishing the reproducibility of the work on "Principal Feature
4    Visualisation in Convolutional Neural Networks", a paper in ECCV 2020.

5    **Scope of Reproducibility**

6    We validated the reproducibility of the work on "Principal Feature Visualisation in Convolutional Neural Networks"[1].
7    We experimented this technique on various images as well as various CNN pre-trained models.

8    **Methodology**

9    We used the authors code and re-implemented it on various images to validated the 4 major claims of the paper. We
10   changed the pre-trained models in the code and noted the best for the tested image classes amongst them.

11   **Results**

12   We used the authors code and implemented it on the images used by the authors as well as other image classes. We
13   compared the results with the Grad-Cam approach and validated all the claims made by authors in the paper.

14   **What was easy**

15   Implementing the code was easy and required very less computational power. The comments with the code by the
16   authors made it easy to comprehend the code and conduct various experiment with it.

17   **What was difficult**

18   The description provided in the paper was difficult to comprehend in the terms of reproducing the code.

19   **Communication with original authors**

20   We contacted an author of the paper. We wanted to understand the reasons about the colours appearing on the result
21   images and in understanding the reason behind the results not appearing appropriately with the inception-v3 model.

# 1    Introduction

As AI has increasingly established itself across technologies and domains, the need for comprehending Machine Learning systems has increased. Convolutional Neural Networks find extensive use in various Machine Learning applications where the data is largely image based. There has been some previous work in visualising and understanding the results by CNNs like Class Activation Mapping, Gradient-based methods. But the authors of paper[1] observed that although Class Activation Mapping is a computationally efficient way to show the support of a class in the input image, but the resulting heatmap is quite coarse.

Also, Gradient-based methods like [3] give a more localised response, but require backpropagation through the whole network, and is very sensitive to edges and noise in the input image. All these methods operate in a supervised manner on one category or feature at a time. As opposed to these, the method in [1] is unsupervised and visualise several categories or features in one pass. It can be applied directly to any bottleneck network without any additional instrumentation. With this reproducibility study, we aim to validate the claims by the authors as stated in paper[1].

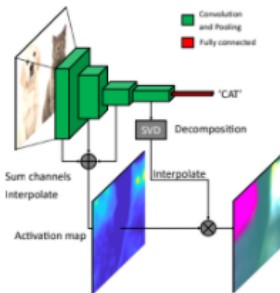

Figure 1: Overview of Principal Feature Visualisation (PFV)method.

# 2    Scope of reproducibility

The paper states 4 major claims around which we have centred the reproudcibility results.

We intend to test the 4 major claims of the paper:

- Contrast: Per-pixel visualisation of the principal contrasting features.
- Lightweight: Requires a single forward pass of the original unmodified network, using only intermediate feature maps.
- Easy to interpret: suppresses non-relevant features.
- Unsupervised: No additional input or prior knowledge about image classes is required.

All of the claims are central to the contribution by the paper. The paper introduces a new visualisation technique for CNNs called Principal Feature Visualisation (PFV). It uses a single forward pass of the original network to map principal features from the final convolutional layer to the original image space as RGB. The paper also states that the stated approach is better than the Grad Cam approach and is faster. To establish these results we ran the Grad Cam code on the same images and compared the the results produced.

# 3    Methodology

We used the code given by authors as it was difficult to reproduce the same code by just reading the paper. Code is written in python PyTorch framework. Implementation of the code was not much of a tedious task as comments available in the code made it easy to understand and modify. We verified various claims made by the authors along with testing the model on diverse set of images in order to understand whether this technique is limited to particular set or not. Moreover, we modified the code to test this technique on various kinds of CNN(Convolutional Neural Networks) models and drawn a comparison between Principal Feature Visualization and Grad-Cam(widely used visualization technique).

### 3.1 Model descriptions

The underlying concept of Principal Feature Visualization is decomposing a feature map into its principal contrasting features for a batch of images. This is accomplished by extracting principal components through singular value decomposition. The decomposed feature map is then interpolated back to the original image space, where we use the activation maps in the preceding layers as spatial weighting. An overview of the method is shown in introduction section.

Detailed mathematics of this technique is covered in the original paper.

### 3.2 Datasets

We majorly used the Open image dataset v6, images by the author provided with the code and some images online. Credits have been mentioned in the GitHub repository.

### 3.3 Hyperparameters

There were as such no use of hyperparameters since its an visualization technique but in order to try this on various models we modified some preprocessing steps. For example, Images to be tested are of shape (3 x H x W), where H and W are height and width respectively and 3 represent RGB channel. So for Inception networks H and W are expected to be at least 299.

### 3.4 Experimental setup

We have used Google Colab free GPU system to run the entire experiment which is Nvidia K80 with 12GB GPU memory and 12 GB of RAM. Entire code is available at https://github.com/abhinav0000004/Principal-Feature-Visualization along with the details of implementation steps.

### 3.5 Computational requirements

As mentioned we used Google Colab free GPU to run the entire setup. Colab gives you a decent GPU for free, which you can continuously run for 12 hours. Google Colab provides Nvidia K80/T4 GPU with 12/16GB of GPU memory having GPU Memory Clock 0.82GHz/1.59GHz along with 2 CPU cores and RAM of 12 GB(which can be upgraded to 26GB).

## 4 Results

Following are the verified results:

- As an output we managed to get per-pixel visualisation of the principal contrasting features.

- It requires only the single pass to interpret the results and much faster than the existing techniques like Grad Cam.

- It managed to identify the strong features of an image.

- Prior information about label class is not required

### 4.1 Result 1

We tried to run this experiment over various diverse images taken from Open Images Dataset v6 and some online images and here is how model with pre-trained weight of VGG-16 performed.

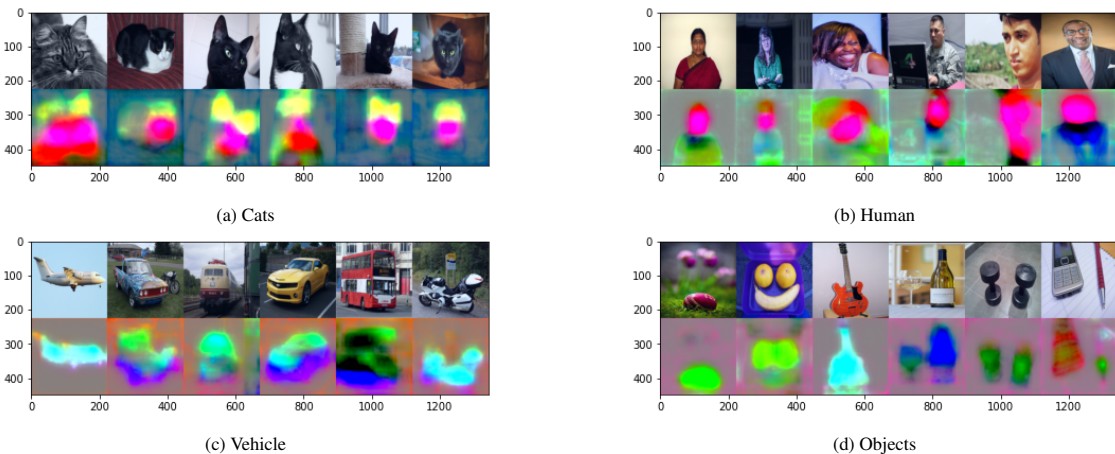

(a) Cats

(b) Human

(c) Vehicle

(d) Objects

Figure 2: Various Image sets

## 4.2 Result 2

These are the results for experimentation with different pre-trained models with the Principal Feature Visualisation technique.

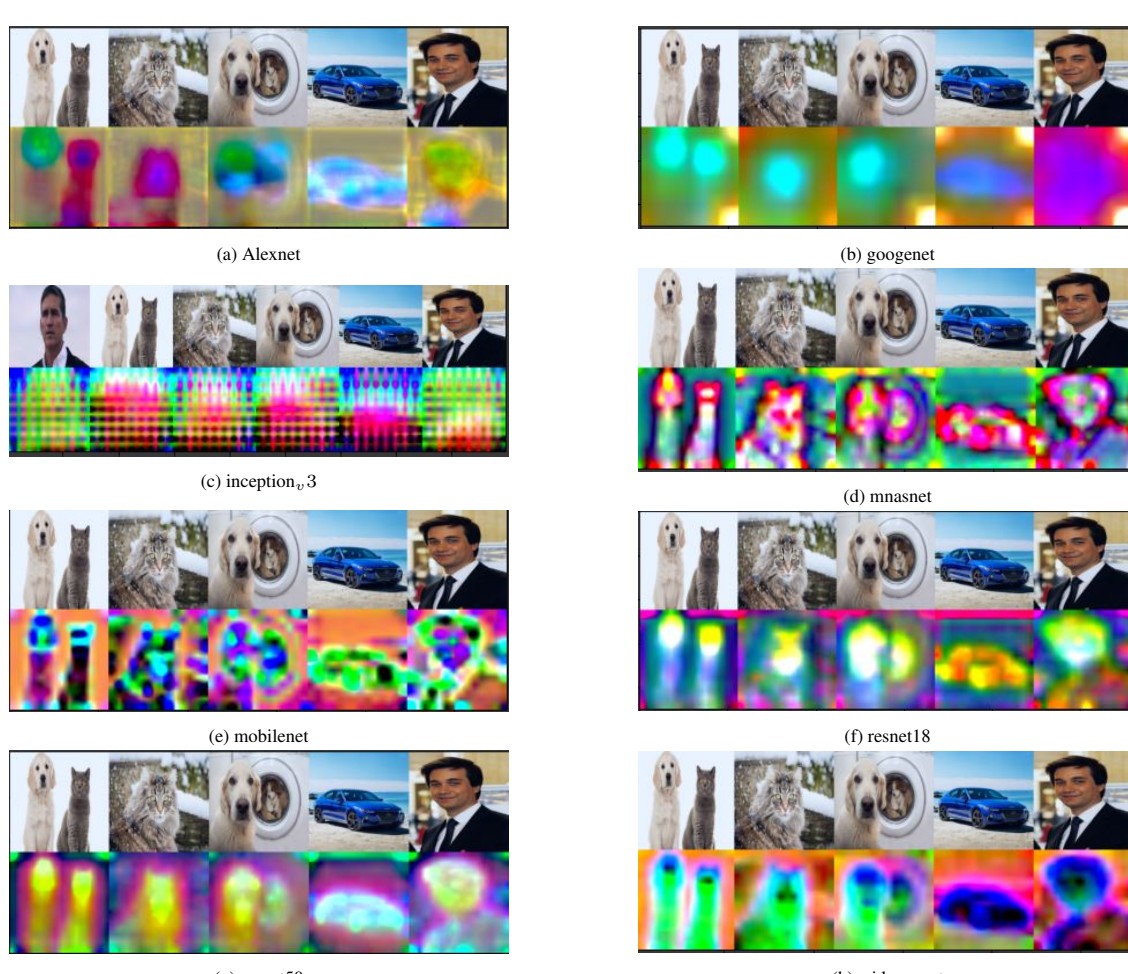

(a) Alexnet

(b) googenet

(c) inception$_v$3

(d) mnasnet

(e) mobilenet

(f) resnet18

(g) resnet50

(h) wide resnet

Figure 3: Various CNN models

### 4.3 Result 3

This section contains the comparison of the results between the Principle feature visualisation and Grad Cam.

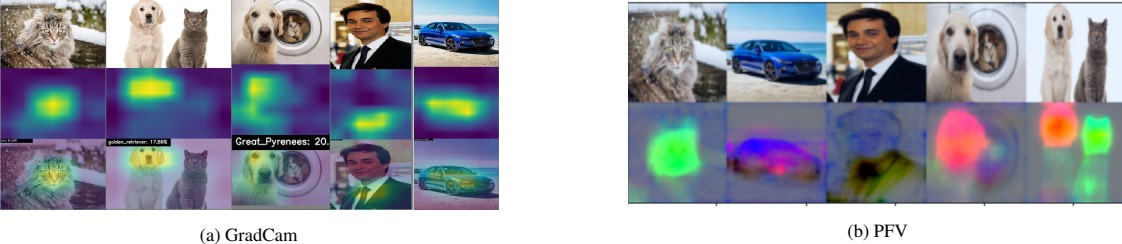

(a) GradCam             (b) PFV

Figure 4: Comparison between GradCam and PFV

### 4.4 Result 4

This section contains the results on objects like table,chair,sofa,etc where this technique failed to visualise the objects correctly.

Credits to images can be found on the GitHub repository that we have created for the results.

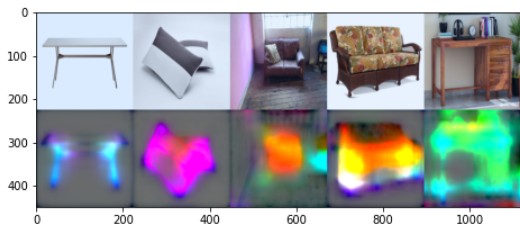

## 5 Discussion

After the complete analysis of the results obtained from various experiments performed on this technique, we believe that all the claims made by the authors are valid. This process has many advantages over existing visualization techniques which include time efficiency, less complexity, easily interpretable, and it doesn't require any knowledge of the class. We even tried various other models that are not mentioned in the original paper and this technique is working well on all those models as well. We believe that this technology can be beneficial in the visualization of the model as well as in transfer learning to choose the correct pre-trained models. Apart from this this techniques fails to correctly visualise the results in case of images of tables, chairs, sofa, pillow.

### 5.1 What was easy

The author's code is clearly written that contains comments wherever required and is easy to run, so it was easy to verify the majority of original claims. Along with that, this code doesn't require much computational power. With just changes in few lines, one can easily perform this technique on various models and diverse dataset. Even if someone is not able to understand the underlying concept of this technique,they can easily use this code.

### 5.2 What was difficult

The underlying concept required a bit of advanced knowledge in the CNN architectures and how they work. Apart from that, it is very difficult to reproduce the code just by reading the paper.

### 5.3 Communication with original authors

We managed to contact one of the author through LinkedIn and clarified the following doubts:

**Doubt 1:** We wanted to understand the reason for different colours that are appearing in the visualizations.

**Author's clarification:** "The three colors represent the top 3 principal components describing the variance in the network's response to a batch of images. The first principal component is red, the second is green and third is blue. Depending on the batch, these three components will describe different things. Also depending on how the network is trained of course.

If the batch contains of all dogs, but different breeds, one can expect the principal components to find what is different between the breeds. For instance that some races have different snouts or ears than others. So in this case, the principal contrasting features which separate the dogs can for instance be the difference in their ears.

If the batch contains a diverse set of images, the principal contrasting components appear to more general similarities and differences. In the paper we construct a batch of images with all the classes in Pascal, and we notice a trend in which animals become yellow, while cars, bicycles and motorcycles become another color. Indicating a similar feature set for these classes. For classes not in Imagenet, which the network is pretrained on, the features a very weak or non-informative.

Remember than the original color mapping is RGB, so in many cases you end up with blended colors, meaning that a class cannot be simply represented as a single principal component. We are currently working on ideas to make the colors more interpretable in general."

**Doubt 2:** We wanted to understand the reason behind such distorted results for the $inception_v3 model$.

**Author's Clarification:** "This is related to skip connections and multiple paths through the network. Inception models have layers with multiple paths through it. In these cases one has to think about where to extract the features and how to go back to the original image space.

In inception networks you can have multiple different pooling operations in each layer. I would therefore think that the artefacts you are observing are due to a wrong unpooling."

Author was very supportive and helpful during the entire period.

# 6 Code Repository of our experiments

We have published the results of our reproducibility study in a GitHub repository: https://github.com/abhinav0000004/Principal-Feature-Visualization

# References

[1]Marianne Bakken1, Johannes Kvam1, Alexey A. Stepanov1, and Asbjørn Berge1 ,"Principal Feature Visualisation in Convolutional Neural Networks"

[2]https://www.pyimagesearch.com/2020/03/09/grad-cam-visualize-class-activation-maps-with-keras-tensorflow-and-deep-learning/

