# OpenReview forum: "Principle Feature Visualisation in Convolutional Neural Networks"
_ML_Reproducibility_Challenge/2020 — Reject_

### Official Review · AnonReviewer3 · 2021-02-17
**Principle Feature Visualisation tool**

**Rating:** 7
**Confidence:** 5

**Review:**

The report is interesting. I want to know how this approach is different than the approach mentioned in Keras-vis (https://github.com/raghakot/keras-vis). A comparative analysis with the Keras-vis approach will be beneficial. It is also required to show some failed cases as well.

**Familiar With The Original Paper:**

I have read the original paper

**Reproducibility Summary:**

Report has summary

---

### Official Review · AnonReviewer2 · 2021-02-28
**Good reproducibility report but needs additional investigation to improve interpretation of results**

**Rating:** 4
**Confidence:** 4

**Review:**

The authors do a commendable job in reporting the state of reproducibility of the original paper and provide a concise summary of their findings. However, I feel that the current report can be significantly improved with additional experiments to further understand the results obtained by the authors. I have presented a detailed evaluation of the reproducibility report with the associated metric below.

Reproducibility report: The authors present a comprehensive summary that outlines their results and contribution accurately.

Scope of reproducibility: The authors present a clear enumeration of the reproducibility scope, that is derived directly from the original paper.

Code: The authors reused the code from the original authors, citing difficulty in writing the code from scratch from the paper itself. However, they fail to provide explicit details of the issues they faced. It would be helpful if they could outline the problems they faced or propose possible additions to the original paper that can be added to the supplementary information in order to facilitate replication by the community. The github repo submitted by the authors is well-organized and serves as a good codebase to reproduce the results of this paper.

Communication with original authors: The authors present a clarification except citing their proactiveness to contact the original authors and clarify the discrepancy in their obtained results. They also present a discussion pertaining to the colors in the visualization results. However, it is unclear if the authors experimented with different batch sizes or having the same image with different images in the batch to observe if the visualization results changed, as alluded to by the original author.

Hyperparameter Search: I feel that the authors lacked in performing a hyperparameter search and evaluating the robustness of the results. Given that the results are susceptible to the statistics of input batch, I would have liked to see experiments to study the impact of batch size or different seed (allowing for different images in the batch) on the visualization results for each image. This issue stands as a significant weakness of the report and therefore the robustness of the results of the original paper remain undetermined.

Ablation Study: The authors' description of the original methodology is not convincing enough and thus it is difficult to comment if the authors understood the original methodology well. The absence of ablation experiments further raise questions about this issue. It would be helpful if the authors could explore some ablation experiments to comment about the different aspects of PFV functioning. For instance, what is the role of the element-wise multiplication with the activation map? Does it help in making the PFVs more focal?

Discussion on results: The authors were able to explore the scope of reproducibility mentioned earlier in their report. However, the dicussion is not very clear. Specifically, what parts of the paper did they feel lacked clarity thus preventing them from writing the code from scratch? Also, how did the code base from the original authors help clarify those ambiguities? The current report lacks details beyond mentioning "required a bit of advanced knowledge in the CNN architectures". I believe adding this detail is crucial to understand the drawbacks of the original paper with respect to its reproducibility.

Recommendations for reproducibility: The authors provide no additional recommendation to the original paper authors based on their experience. A good reproducibility report often entails specific recommendations that could help make the original paper easier to understand and reproduce.

Results beyond the paper: The authors perform additional experiments using different model architectures. I feel this is very helpful and extremely commendable. However, they do not provide further insight into the results. For instance, why does PFV work fine for Resnet but not InceptionNet even though both of them consist of skip connections (which is cited as the drawback of PFV). Similarly, I am not clear why PFV fails to visualize objects correctly. The results seem to portray that PFV was able to locate these objects in Section 4.4. These ambiguities in the paper need to be resolved and currently serve as severe drawbacks. Furthermore, did the authors explore visualization results for images where more than one entity in present? Is PFV, being a class agnostic unsupervised method, able to identify all entities in the image or does it focus on one of them? Does this result change with changing batch statistics?

Overall organization and clarity: I found some grammatical errors and typos in the paper. I believe the overall presentation of the paper could be improved as well.

Overall, I think the authors have made a good preliminary attempt at replicating the paper and I hope this experience has provided them valuable insights into the method and the field in general. However, the report in its current state has severe drawbacks as mentioned above. If the authors can address these concerns, I am happy to change my evaluation.

**Familiar With The Original Paper:**

I have read the original paper

**Reproducibility Summary:**

Report has summary

---

### Official Review · AnonReviewer1 · 2021-03-02
**The reproducibility report builds on existing code and conducts a preliminary experimental study on a small set of images, which does not suffice to validate original claims.**

**Rating:** 4
**Confidence:** 4

**Review:**

The paper reproduces Principal Feature Visualisation for CNNs by reusing the provided code of the original authors and running it on a chosen set of images. The authors build their study on four claims extracted from the original paper (contrast, lightweight, ease of interpretability, unsupervised) and use a dataset comprising images from Open image dataset v6, example images of the code repository and "some images online". The authors state the the four claims can be verified.

While the paper shows interesting results of running the reused code on the chosen set of images, the paper neither critically assesses the implementation of the approach nor follows a structured approach to reproduce experimental outcomes and findings.

The original paper evaluates the approach for the debugging classification errors (for dog breeds) and transfer learning (using the Pascal VOC2012 dataset for fine-tuning). It would be highly interesting to design and conduct experiments for other target classes / target datasets or use other CNN architectures. While the authors actually vary the latter (and use established pre-trained models, such as AlexNet or MobileNet), they only show the results on five images and do not further assess the applicability for debugging or transfer learning.

**Familiar With The Original Paper:**

I have read the original paper

**Reproducibility Summary:**

Report has summary

---

### Decision · Program_Chairs · 2021-03-31

**Decision:**

Reject

**Comment:**

Overall reviews and/or the paper content not good enough for the AC to recommend to the journal.